# Automated Machine Learning (AutoML)-Derived Preconception Predictive Risk Model to Guide Early Intervention for Gestational Diabetes Mellitus

**DOI:** 10.3390/ijerph19116792

**Published:** 2022-06-01

**Authors:** Mukkesh Kumar, Li Ting Ang, Hang Png, Maisie Ng, Karen Tan, See Ling Loy, Kok Hian Tan, Jerry Kok Yen Chan, Keith M. Godfrey, Shiao-yng Chan, Yap Seng Chong, Johan G. Eriksson, Mengling Feng, Neerja Karnani

**Affiliations:** 1Singapore Institute for Clinical Sciences, Agency for Science Technology and Research, Singapore 138632, Singapore; mukkesh_kumar@sics.a-star.edu.sg (M.K.); ang_li_ting@sics.a-star.edu.sg (L.T.A.); png_hang@sics.a-star.edu.sg (H.P.); maisie_ng@sics.a-star.edu.sg (M.N.); karen_tan@sics.a-star.edu.sg (K.T.); obgchan@nus.edu.sg (S.-y.C.); obgcys@nus.edu.sg (Y.S.C.); obgjge@nus.edu.sg (J.G.E.); 2Bioinformatics Institute, Agency for Science Technology and Research, Singapore 138632, Singapore; 3Saw Swee Hock School of Public Health, National University of Singapore, National University Health System, Singapore 119077, Singapore; 4Obstetrics & Gynaecology Academic Clinical Programme, Duke-NUS Medical School, Singapore 169857, Singapore; loyseeling@duke-nus.edu.sg (S.L.L.); tan.kok.hian@singhealth.com.sg (K.H.T.); jerrychan@duke-nus.edu.sg (J.K.Y.C.); 5Department of Reproductive Medicine, KK Women’s and Children’s Hospital, Singapore 229899, Singapore; 6Division of Obstetrics and Gynecology, KK Women’s and Children’s Hospital, Singapore 229899, Singapore; 7Cancer and Stem Cell Biology, Duke-NUS Medical School, Singapore 169857, Singapore; 8Human Potential Translational Research Programme, Department of Obstetrics and Gynaecology, Yong Loo Lin School of Medicine, National University of Singapore, Singapore 119077, Singapore; 9MRC Lifecourse Epidemiology Centre, NIHR Southampton Biomedical Research Centre, University Hospital Southampton NHS Foundation Trust, University of Southampton, Southampton SO17 1BJ, UK; kmg@mrc.soton.ac.uk; 10Department of General Practice and Primary Health Care, University of Helsinki, 00100 Helsinki, Finland; 11Folkhälsan Research Center, 00250 Helsinki, Finland; 12Institute of Data Science, National University of Singapore, Singapore 119077, Singapore; 13Department of Biochemistry, Yong Loo Lin School of Medicine, National University of Singapore, Singapore 119077, Singapore

**Keywords:** Asian populations, digital health, gestational diabetes mellitus, HbA_1c_, machine learning, preconception care, prediction, preterm birth, public health, risk factors

## Abstract

The increasing prevalence of gestational diabetes mellitus (GDM) is contributing to the rising global burden of type 2 diabetes (T2D) and intergenerational cycle of chronic metabolic disorders. Primary lifestyle interventions to manage GDM, including second trimester dietary and exercise guidance, have met with limited success due to late implementation, poor adherence and generic guidelines. In this study, we aimed to build a preconception-based GDM predictor to enable early intervention. We also assessed the associations of top predictors with GDM and adverse birth outcomes. Our evolutionary algorithm-based automated machine learning (AutoML) model was implemented with data from 222 Asian multi-ethnic women in a preconception cohort study, Singapore Preconception Study of Long-Term Maternal and Child Outcomes (S-PRESTO). A stacked ensemble model with a gradient boosting classifier and linear support vector machine classifier (stochastic gradient descent training) was derived using genetic programming, achieving an excellent AUC of 0.93 based on four features (glycated hemoglobin A_1c_ (HbA_1c_), mean arterial blood pressure, fasting insulin, triglycerides/HDL ratio). The results of multivariate logistic regression model showed that each 1 mmol/mol increase in preconception HbA_1c_ was positively associated with increased risks of GDM (*p* = 0.001, odds ratio (95% CI) 1.34 (1.13–1.60)) and preterm birth (*p* = 0.011, odds ratio 1.63 (1.12–2.38)). Optimal control of preconception HbA_1c_ may aid in preventing GDM and reducing the incidence of preterm birth. Our trained predictor has been deployed as a web application that can be easily employed in GDM intervention programs, prior to conception.

## 1. Introduction

The prevalence of gestational diabetes mellitus (GDM) is increasing globally, affecting one in five pregnancies in some populations [1]. GDM is a condition in which a woman without previous diabetes develops glucose intolerance during pregnancy [2]. This condition increases the risk of developing GDM-related complications such as hypertensive disorders of pregnancy, fetal macrosomia, caesarean section, shoulder dystocia and birth injuries [3]. Poorly controlled GDM also increases risks of premature birth, perinatal mortality and neonatal morbidity. GDM has long-term implications as women with a history of GDM have a 10-fold higher risk of developing type 2 diabetes (T2D) as well as higher risk of developing cardiovascular adversities compared to those with a normoglycemic pregnancy [4,5]. Offspring of mothers with GDM are also at an increased risk of having cardiometabolic adversities, resulting in a transgenerational cycle of diabetes and cardiovascular diseases [6].

Healthcare systems across the world use either the high risk selective screening approach or universal screening of GDM in pregnant women. The American Diabetes Association (ADA) endorses the use of either a one-step approach (IADPSG diagnostic criteria, fasting two-hour, three-point 75 g oral glucose tolerance test (OGTT)) or an older two-step approach (non-fasting one-hour 50 g glucose challenge test (GCT), followed by diagnostic fasting three-hour 100 g OGTT on a subset of women exceeding the glucose threshold value of GCT) at 24–28 weeks’ gestation [7]. The UK NICE recommends high risk selective screening for women with known GDM risk factors, such as obesity (body mass index (BMI) ≥30 kg/m^2^), family history of diabetes, history of GDM, previous delivery of a macrosomic baby (≥4.5 kg) and being in an ethnic group with a high prevalence of diabetes (South Asian, Black Caribbean or Middle Eastern) [8]. In the latest UK NICE 2015 guidelines, women with a history of GDM are offered an OGTT at their booking appointment [8]. Women with other risk factors are offered an OGTT at 24–28 weeks’ gestation. The International Diabetes Federation (IDF) GDM Model of Care [9] recommends that all pregnant women are screened at first visit by a fasting glucose, HbA_1c_ or random glucose sample to rule out pre-existing diabetes. In those with normal early screening, an OGTT is performed at 24–28 weeks’ and 32 weeks’ gestation (for high risk women) to assess the risk of GDM.

Pre-existing abnormalities in maternal metabolism are important factors in the pathophysiology of metabolic diseases and fetal programming. GDM intervention typically focuses on counseling, dietary modification and increasing physical activity. The daily self-monitoring of blood glucose is aimed at normalizing blood glucose levels and reducing the complications of GDM. Primary lifestyle interventions to manage GDM, such as diet and exercise in the second trimester, provide limited benefits for the mother and child due to late implementation, poor adherence and generic guidelines [10]. Preconception presents an important opportunity to break the intergenerational cycle of chronic metabolic disorders. The Lancet series on preconception maternal health in 2018 highlighted preconception as a critical period for shaping pregnancy outcomes and subsequent maternal and child health [11,12,13].

In recent years, some machine learning models have been developed for population based GDM risk stratification. However, the current state-of-the-art models are only applicable during pregnancy, which can be too late for effective intervention. Artzi et al. trained a LightGBM gradient boosting classifier with Israel’s Electronic Health Records (EHR) data for onset of GDM (area under the receiver operating characteristic curve (AUC) of 0.80 was achieved with nine questionnaire features) [14]. In another study, Wu et al. trained a logistic regression classifier with China’s EHR data for early GDM prediction (AUC of 0.77 was achieved with seven clinical features) [15]. To date, there have been no studies applying machine learning for GDM risk assessment in a preconception population. We therefore would like to suggest a paradigm shift in GDM management strategy.

In this study, we developed a machine learning model for early prediction of GDM during preconception among women in Singapore. Taking a longitudinal approach, we also assessed the associations of the strongest predictors with GDM and adverse birth outcomes (preterm birth, low birthweight at term and large for gestational age infant). Our machine learning models were implemented using data from the prospective Singapore Preconception Study of Long-Term Maternal and Child Outcomes (S-PRESTO) cohort study.

## 2. Materials and Methods

### 2.1. Study Design

S-PRESTO (ClinicalTrials.gov NCT03531658) is a prospective, preconception cohort study of multi-ethnic groups (Chinese, Malay, Indian or any combination of these three ethnicities) [16]. Women planning for pregnancies were recruited from the KK Women’s and Children’s Hospital (KKH) and community between February 2015 and October 2017. There were 1032 unique participants for preconception; 475 conceived singleton pregnancies within a year of enrollment into the study, and 373 remained in the study and had a livebirth. The mother–child dyads have been followed for seven years, with longitudinal phenotypic data collected across multiple health domains.

Maternal glucose tolerance status was assessed longitudinally using 75 g 2 h oral glucose tolerance test (OGTT) preconception, mid-gestation (median 28.1 weeks, interquartile range 27.3–28.7 weeks) and 3 months postpartum, alongside glycated hemoglobin A_1c_ (HbA_1c_) at the same timepoints. The International Association of Diabetes and Pregnancy Study (IADPSG)/World Health Organization (WHO) 2013 criteria (fasting plasma glucose ≥ 5.1–6.9 mmol/L, 1 h plasma glucose ≥ 10.0 mmol/L and 2 h plasma glucose ≥ 8.5–11.0 mmol/L) were used to diagnose GDM [17]. The WHO 2006 criteria (fasting plasma glucose ≥ 7.0 mmol/L or 2 h plasma glucose ≥ 11.1 mmol/L) were used to diagnose impaired fasting glucose (IFG), impaired glucose tolerance (IGT) and type 2 diabetes (T2D) [18]. An HbA_1c_ of ≥6.5% was used as the cut-off point for diagnosing diabetes based on WHO recommendations [19].

Participants diagnosed with T2D based on preconception and 3 months postpartum OGTT or HbA_1c_ readings were excluded from model training. GDM analysis was restricted to mothers whose gestation at the time of antenatal OGTT was 24^+1^–28^+6^ weeks (gestational age is given as weeks^+days^). Participants of mixed ethnicity or unclassifiable GDM status due to missing glucose readings were removed from the final analysis set.

Our models were built using 222 preconception women who had complete data on demographics, medical/obstetric history, physical measures, blood-derived markers, lifestyle factors and antenatal OGTT (Figure 1). The prevalence of GDM was 13.1% in our training dataset. Participant characteristics are presented in Table 1.

Information on demographics (age, ethnicity) and medical/obstetric history (family history of diabetes mellitus, history of GDM, parity and medical history of high blood pressure) were derived from preconception questionnaires. Lifestyle factors on self-reported smoking and alcohol consumption were also collected at preconception.

The physical measures at preconception were included for feature selection modeling. Weight was measured to the nearest 0.1 kg (SECA 803) and height to the nearest 0.1 cm (SECA 213). BMI was derived using weight divided by height squared (kg/m^2^). Waist circumference was measured to the nearest 0.1 cm (SECA 203). Additionally, mid-upper arm circumference was measured to the nearest 0.1 cm, midway between acromion process and olecranon process (SECA 212). Systolic and diastolic blood pressure were measured using the Microlife BP 3AS1-2 blood pressure device. Mean arterial blood pressure was further derived by doubling the diastolic blood pressure and adding to the systolic blood pressure, with the composite sum divided by 3.

Sodium fluoride/potassium oxalate tubes were used to collect blood samples for plasma glucose measurement. Potassium EDTA tubes were used to collect whole blood samples for HbA_1c_ measurement. All samples were kept at 4 °C, immediately sent to the hospital laboratory, centrifuged within 30 min and analyzed within 1 h from the time of earliest blood draw. Fasting plasma glucose, 30 min plasma glucose, 1 h plasma glucose (antenatal OGTT only), 90 min plasma glucose (antenatal OGTT only), 2 h plasma glucose and HbA_1c_ concentrations were measured using the ARCHITECT c8000 Clinical Chemistry Analyzer (Abbott Laboratories), which is a National Glycohemoglobin Standardisation Program (NGSP) certified method for HbA_1c_ testing. The preconception HbA_1c_ marker was included for feature selection modeling.

Longitudinally obtained plasma samples were analyzed for fasting insulin, triglycerides (TGs), high density lipoprotein (HDL) cholesterol and gamma-glutamyl transferase at the National University Hospital (NUH) clinical laboratory (accredited by the College of American Pathologists [20]). Maternal venous blood was collected into silicone coated tubes, and serum was obtained by centrifugation at 1600× *g* for 10 min at 4 °C. The serum was stored at −80 °C until sample batch analysis. Insulin was measured using the Sandwich immunoassay (Beckman DxI 800 analyzer, manufactured by Beckman Coulter in Fullerton, CA, USA). Using a Beckman AU5800 analyzer, TG and gamma-glutamyl transferase were measured by colorimetric assays and HDL cholesterol using an enzymatic assay. These blood markers were subsequently used for the derivation of metabolic indices and machine learning modeling.

The homeostasis model assessment of insulin resistance (*HOMA-IR*) index was calculated based on the formula [21]: $$HOMA-IR=\frac{fasting\;glucose\;in\frac{mmol}{L}\times fasting\;insulin\;in\frac{mU}{L}}{22.5}$$

In addition, the TG/HDL cholesterol ratio was calculated based on the fasting lipid concentrations to assess insulin resistance [22].

Fatty liver index as a surrogate marker of non-alcoholic fatty liver disease (NAFLD) was calculated with 4 variables (triglycerides (*TG*s), *BMI*, gamma-glutamyl transferase (*GGT*) and waist circumference (*WC*)) [23]:$$FLI=\frac{\left(e^{0.953\times log_{e}\left(TG\right)+0.139\times BMI+0.718\times log_{e}\left(GGT\right)+0.053\times WC-15.745}\right)}{\left(1+e^{0.953\times log_{e}\left(TG\right)+0.139\times BMI+0.718\times log_{e}\left(GGT\right)+0.053\times WC-15.745}\right)}\times 100$$

Metabolic syndrome status was defined when three or more of the following criteria were fulfilled: waist circumference > 80 cm, triglycerides ≥ 1.7 mmol/L, HDL cholesterol ≤ 1.3 mmol/L, blood pressure ≥ 130/85 mm Hg, fasting plasma glucose ≥ 6.1 mmol/L [24].

Age, ethnicity, family history of diabetes mellitus, history of GDM, parity, height, BMI, mid-upper arm circumference, mean arterial blood pressure, HbA_1c_, fasting insulin, self-reported smoking, self-reported alcohol consumption, TG/HDL ratio, fatty liver index and metabolic syndrome variables were included for feature selection modeling.

### 2.2. Machine Learning Methodology and Statistical Analyses

Our methodological novelty lies in combining coalitional game theory concepts with evolutionary algorithm-based automated machine learning (AutoML). Automating the process of machine learning enables the best possible model to be built for our supervised machine learning problem. The optimal machine learning pipelines were automatically generated using genetic programming (GP), a type of evolutionary algorithm [25,26]. An introduction to GP is provided in the Appendix A. In brief, GP solves machine learning tasks based on random mutation, crossover, fitness functions and generations to arrive at optimal solutions (models and hyperparameters).

The Shapley additive explanations (SHAP) framework [27] was combined with the evolutionary algorithm-based Tree-Based Pipeline Optimization Tool (TPOT) [28] to discover novel features and select optimal supervised machine learning models. We explored the interaction effects of multiple predictors using the SHAP framework methodology. In game theory, the Shapley value is the average expected marginal contribution of one player across all possible permutations of players (average effects of team member composition and team size). The Shapley value helps to determine a payoff for all the game players when each player might have contributed more or less than the others when working in coalition. In machine learning, game players are the features, and collective payout is the model prediction. The SHAP framework provides local explanations based on exact Shapley values to understand the global model structure. For every possible feature ordering, features are introduced one at a time into a conditional expectation function of the model’s output, and changes in expectation are attributed to the introduced feature, averaged over all possible feature orderings in a fair manner. SHAP values represent a change in log odds ratio. Lundberg and Lee have proposed SHAP as the only additive feature attribution method that satisfies two important properties of game theory—additivity (local accuracy) and monotonicity (consistency) [27]. The integrated game theoretical approach with automated machine learning further advances biomedical data science for data-driven precision care. 

The AutoML models were built using Anaconda’s distribution of Python v3.7.9 programming language in a JupyterLab computational environment. Community-developed Python packages were used for modular programming: Pandas v0.25.3, Numpy v1.19.2, Matplotlib v3.3.2, Scikit-learn v0.23.2, TPOT v0.11.7 and Shap v0.37.0. We trained the AutoML models on a Linux server with an Intel Xeon Gold 6138 CPU processor. In the TPOT classifier, the search for optimal machine learning pipelines was run over 100 generations with 100 individuals retained in the genetic programming population of every generation. We used 5-fold stratified cross validation to preserve the same proportion of GDM cases in each fold, and model performances were evaluated using the area under the receiver operating characteristic curve (AUC). 

The AutoML feature selection model based on preconception feature variables was trained with GDM as the outcome; top predictors with SHAP value magnitudes greater than zero were included in the GDM prediction models. Sensitivity analyses were performed to explore the prediction effects of fasting glucose, systolic blood pressure and HOMA-IR index in the proposed AutoML model. We also assessed the associations between the strongest predictors and GDM outcome/adverse birth outcomes (preterm birth, low birthweight at term and large for gestational age infants). Preterm birth was defined as livebirth before 37 weeks of pregnancy [29]. Low birthweight at term was defined as birthweight less than 2500 g in term births (37–42 weeks of pregnancy) [29]. The sex-specific birthweight for gestational age percentile was derived by making reference to Growing Up in Singapore Towards Healthy Outcomes (GUSTO) healthy newborn weight percentile [30], which was based on the generic reference for birthweight percentiles created by Mikolajczyk et al. [31]. Large for gestational age infants have a birthweight of more than 90th percentile. Additional sensitivity analyses were performed by excluding preconception women with prediabetes (IFG and IGT). All association analyses were performed using Stata/MP 17.0 software (StataCorp LP, College Station, TX, USA).

## 3. Results

### 3.1. Top Predictors from AutoML Feature Selection Model

Figure 2 presents the SHAP global importance plot of the AutoML feature selection model. A stacked ensemble model with a random forest classifier and linear support vector machine classifier (stochastic gradient descent training) was the best machine learning pipeline evaluated by TPOT (AUC: 0.89). The top preconception feature variables impacting the model outputs were HbA_1c_, fatty liver index, mean arterial blood pressure, fasting insulin, TG/HDL ratio, height, age, mid-upper arm circumference, BMI, parity, alcohol consumption, family history of diabetes mellitus and Chinese ethnicity. 

Pre-pregnancy BMI demonstrated small predictive effects relative to preconception HbA_1c_. Chinese women also had a higher risk of GDM when compared with Indian and Malay women. The latter observation could be attributed to the high proportion of Chinese ethnic participants in the analysis set (79.3%). A history of GDM was a redundant feature in the AutoML feature selection model possibly due to the low frequency of participants with a history of documented GDM (2.7%). Metabolic syndrome status preconception did not contribute to GDM prediction. 

### 3.2. Preconception Predictive Risk Model

The preconception predictive risk model for GDM was sequentially constructed using top predictors with SHAP value magnitudes greater than zero (Table 2). Preconception HbA_1c_ alone was able to predict GDM outcome with high discrimination (AUC: 0.81). A model with nine features obtainable non-invasively (mean arterial blood pressure, height, age, mid-upper arm circumference, BMI, parity, alcohol consumption, family history of diabetes, Chinese ethnicity) was also able to predict GDM outcome with good performance (AUC: 0.81). The optimal machine learning pipeline comprises five features (HbA_1c_, fatty liver index, mean arterial blood pressure, fasting insulin, TG/HDL ratio). The extra trees classifier was the best machine learning pipeline evaluated by TPOT (AUC: 0.93). In the sensitivity analysis (see Appendix A), we observed that model performance was still maintained by dropping the fatty liver index as a feature variable. Based on the remaining four features, a stacked ensemble model with a gradient boosting classifier and linear support vector machine classifier (stochastic gradient descent training) was the best machine learning pipeline evaluated by TPOT (AUC: 0.93). The four-feature model comprising HbA_1c_, mean arterial blood pressure, fasting insulin and TG/HDL ratio is our proposed solution for a preconception-based GDM predictor. The exported AutoML pipeline for the best predictive model is provided in the Appendix A.

### 3.3. Associations of Top Predictors and GDM Outcome 

Table 3 presents the associations of the strongest predictors identified from the AutoML feature selection model for GDM. Each 1 mmol/mol increase in preconception HbA_1c_ was positively associated with GDM, independent of maternal ethnicity, age, parity, family history of diabetes mellitus and pre-pregnancy BMI (*p* = 0.001, OR (95% CI) 1.34 (1.13–1.60)).

### 3.4. Associations of Top Predictors and Adverse Birth Outcomes (Preterm Birth, Low Birthweight at Term and Large for Gestational Age Infant)

Similarly, Table 4 presents the associations of top GDM predictors with adverse birth outcomes (preterm birth, low birthweight at term and large for gestational age infant). Each 1 mmol/mol increase in preconception HbA_1c_ was positively associated with preterm birth outcome, independent of maternal ethnicity, age, parity, family history of diabetes mellitus, pre-pregnancy BMI, GDM diagnosis, total gestational weight gain and child sex (*p* = 0.011, OR: 1.63 (1.12–2.38)). However, preconception HbA_1c_ was not associated with low birthweight at term (OR: 1.13 (0.86–1.49)) or large for gestational age infant (OR: 1.06 (0.92–1.21)). We additionally found that pre-pregnancy BMI was positively associated with large for gestational age infant (*p* < 0.001, OR: 1.20 (1.10–1.31)).

After excluding women with prediabetes, the associations between preconception HbA_1c_ and a GDM outcome (*p* = 0.003, OR: 1.32 (1.10–1.59)) and with a preterm birth outcome (*p* = 0.010, OR: 1.75 (1.14–2.67)) were not materially changed.

## 4. Discussion

### Primary Findings

We built an effective preconception-based GDM predictor by integrating game theory concepts with evolutionary algorithm-based AutoML. Our proposed AutoML model was derived using genetic programming and achieved an excellent AUC of 0.93 with four features (HbA_1c_, mean arterial blood pressure, fasting insulin, TG/HDL ratio). A stacked ensemble model with the gradient boosting classifier and linear support vector machine classifier (stochastic gradient descent training) was the best machine learning pipeline evaluated by TPOT. The preconception predictive risk model can be leveraged as a risk stratification tool during preconception care to identify Asian women at high risk of developing GDM, enabling early intervention. Alternatively, our non-invasive model trained with nine features (mean arterial blood pressure, height, age, mid-upper arm circumference, BMI, parity, alcohol consumption, family history of diabetes, Chinese ethnicity) provides an alternative for clinical implementation if blood-derived markers are unavailable (AUC: 0.81).

Population-based research on preconception HbA_1c_ and its relationship/association with GDM and adverse birth outcomes remains limited. In our study, HbA_1c_ was the top predictive feature discovered from AutoML feature selection modeling. The physiological variation in HbA_1c_ can be attributed to increased red cell turnover during pregnancy with new erythrocytes exposed to a lower time-averaged glucose concentration [32] and decreasing insulin sensitivity with increasing gestation [33]. 

In the fully adjusted logistic regression model (adjusted for maternal ethnicity, age, parity, family history of diabetes mellitus and pre-pregnancy BMI), preconception HbA_1c_ was associated with increased risks of GDM. Preconception HbA_1c_ alone had a high predictive performance in the AutoML model (AUC: 0.81). Similarly in the sensitivity analyses, the predictive performance of the AutoML model was stronger with preconception HbA_1c_ (AUC: 0.93) than preconception fasting glucose (AUC: 0.87), implying that early GDM risk stratification can be significantly improved with the inclusion of preconception HbA_1c_ over preconception fasting glucose. Moreover, HbA_1c_ offers greater clinical convenience than fasting glucose as there is no fasting requirement, less biological variation and greater pre-analytical stability [34]. As HbA_1c_ is a measure of how glucose has interacted with erythrocytes up to a three-month period [35], our findings suggest that women who develop GDM may have impaired glucose homeostasis prior to pregnancy itself. 

The clinical usefulness of preconception HbA_1c_ can be extended to adverse pregnancy outcomes. In a Swedish study by Ludvigsson et al. [36], women with periconceptional HbA_1c_ levels within recommended target levels (HbA_1c_ < 6.5%) were at increased risk of preterm delivery. The risk of early preterm birth increased with increasing HbA_1c_ levels in normal pregnancies and among women with type 1 diabetes [36]. Our study provides further evidence that preconception HbA_1c_ is an independent risk factor for preterm birth. In the fully adjusted logistic regression model (adjusted for maternal ethnicity, age, parity, family history of diabetes mellitus, pre-pregnancy BMI, GDM diagnosis, total gestational weight gain and child sex), preconception HbA_1c_ was associated with increased risks of preterm birth. Associations between preconception HbA_1c_ and GDM and preterm birth were not materially changed after excluding women with prediabetes, indicating that preconception HbA_1c_ is a reliable marker in predicting GDM/preterm birth even within normal HbA_1c_ range.

Blood pressure changes between preconception and pregnancy are underexplored in the literature. In our study, mean arterial blood pressure feature was another critical component of the AutoML model. Although mean arterial blood pressure at preconception was not associated with GDM outcome, the linkage between preconception blood pressure and physiological changes associated with pregnancy complications warrants further investigation.

The TG/HDL ratio is a surrogate marker for insulin resistance and was one of the top five features in the AutoML feature selection model. GDM is a condition of increased insulin resistance, and this shifts the balance of lipid processing as reflected by the TG/HDL ratio [37,38]. The four features in AutoML modeling for GDM prediction (HbA_1c_, mean arterial blood pressure, fasting insulin and TG/HDL ratio) discovered through genetic programming are suggestive of transient insulin resistance at preconception and reflect the women’s pre-existing metabolic physiology, which clearly has a bearing on the women’s ability to amount an appropriate metabolic adaptation to pregnancy in response to signals from the conceptus to ensure a successful pregnancy. Dysfunctional metabolic adaptation can thus lead to gestational diabetes and preterm birth.

## 5. Limitations

This study has some limitations due to scarcity of longitudinal data. Our AutoML model was trained on a limited S-PRESTO cohort of 222 preconception women. However, the prospective S-PRESTO data capture complex clinical pathways during pregnancy initiation and are less prone to differential measurement errors. A sub-cohort analyses by individual ethnic groups can be trained with larger clinical datasets such as the electronic health records. No replication cohort was available, and our proposed model should be evaluated in confirmatory studies. The four features in AutoML modeling for GDM prediction need to be evaluated in an early pregnancy cohort for generalizability.

### Comparison with Prior Work

The implementation of our GDM risk prediction algorithm during preconception care would enable early engagement of women for preventive intervention, compared to existing pregnancy-based GDM risk prediction algorithms [14,15] developed for antenatal care. In another recent study by Wu et al. [39], an early pregnancy prediction model for GDM was developed based on genetic variants (four genetic susceptible single nucleotide polymorphisms (SNPs)) and six basic clinical features (AUC: 0.73). The latter model requires more advanced laboratory testing for SNPs, which may not be routinely available in all standard clinical laboratories. Xiong et al. [40] developed high performance machine learning models with the linear support vector machine classifier and LightGBM gradient boosting classifier using 10–19 weeks’ gestation data (AUC: 0.91–0.98), which may be too late for effective GDM interventions. With four basic clinical features measured at preconception and high prediction performance of AUC: 0.93, our stacked ensemble model with the gradient boosting classifier and linear support vector machine classifier (stochastic gradient descent training) offers a simpler solution for early GDM prediction.

## 6. Conclusions

Leveraging AI and evolutionary algorithms, we devised a population-based predictive care solution to assess the risk of developing GDM in preconception of Asian women. An optimal control of preconception HbA_1c_ has the potential to lower the risk of GDM and reduce the incidence of preterm birth. Our trained classifier has been deployed in a web application for GDM prevention programs and intervention with early-stage nutritional and lifestyle changes during preconception care. The GDM predictor can also be combined with a digital health intervention such as a smartphone application.

## Figures and Tables

**Figure 1 ijerph-19-06792-f001:**
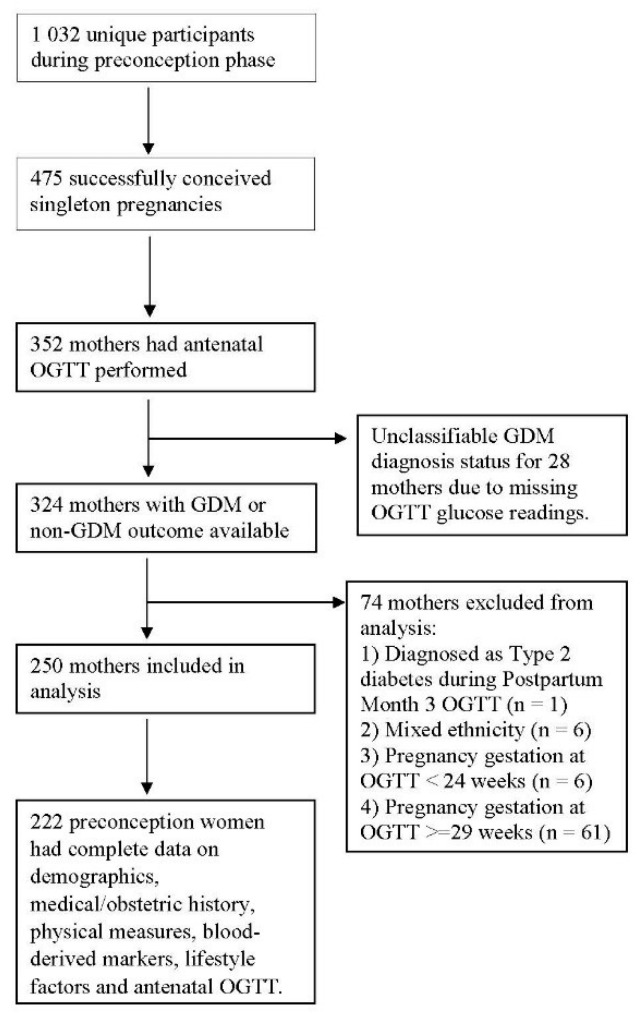
Sample Participant characteristics. Sample selection flowchart of 222 preconception women who had complete data on demographics, medical/obstetric history, physical measures, blood-derived markers, lifestyle factors and antenatal OGTT for machine learning models.

**Figure 2 ijerph-19-06792-f002:**
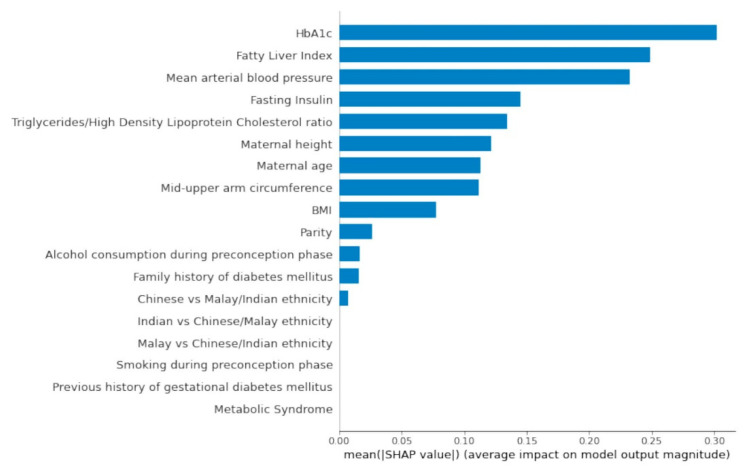
SHAP Global Importance Plot. Global importance of individual features and their correlation with GDM/non-GDM outcomes estimated using the Shapley values computed from coalitional game theory. SHAP values represent a change in log odds ratio. SHAP values of zero means that the feature does not contribute to the prediction.

**Table 1 ijerph-19-06792-t001:** Participant characteristics at preconception baseline. Participant characteristics table on demographics, medical/obstetric history, physical measures, blood-derived markers, lifestyle factors, metabolic indices, prediabetes status, antenatal OGTT and adverse birth outcomes. Continuous variables are presented as group mean value and standard deviation. Categorical variables are presented as count and percentage.

	S-PRESTO (*n* = 222)
**Demographics**	
Age (years), mean ± SD	30.51 ± 3.11
Ethnicity, *n* (%)	
Chinese	176 (79.28)
Malay	30 (13.51)
Indian	16 (7.21)
**Medical/Obstetric History**	
Family history of diabetes mellitus, *n* (%)	
Yes	63 (28.38)
No	159 (71.62)
History of GDM, *n* (%)	
Yes	6 (2.70)
No	216 (97.30)
Parity, *n* (%)	
Nulliparous	140 (63.06)
Multiparous	82 (36.94)
Medical history of high blood pressure, *n* (%)	
Yes	0 (0.00)
No	222 (100.00)
**Physical Measures at Preconception**	
Pre-pregnancy weight (kg), mean ± SD	59.31 ± 11.80
Maternal height (cm), mean ± SD	159.96 ± 5.55
Pre-pregnancy BMI (kg/m2), mean ± SD	23.18 ± 4.52
Waist circumference (cm), mean ± SD	81.35 ± 10.10
Mid-upper arm circumference (cm), mean ± SD	27.27 ± 4.08
Systolic blood pressure (mm Hg), mean ± SD	104.15 ± 8.92
Diastolic blood pressure (mm Hg), mean ± SD	67.38 ± 7.51
Mean arterial blood pressure (mm Hg), mean ± SD	79.63 ± 7.48
**Blood-Derived Markers at Preconception**	
HbA1c (mmol/mol), mean ± SD	31.80 ± 2.73
Fasting glucose (mmol/L), mean ± SD	4.72 ± 0.33
Fasting insulin (mU/L), mean ± SD	5.97 ± 4.83
Triglycerides (mmol/L), mean ± SD	0.81 ± 0.38
High density lipoprotein cholesterol (mmol/L), mean ± SD	1.48 ± 0.28
Gamma-glutamyl transferase (U/L), mean ± SD	18.99 ± 14.28
**Lifestyle Factors at Preconception**	
Self-reported smoking, *n* (%)	
Yes	6 (2.70)
No	216 (97.30)
Self-reported alcohol consumption, *n* (%)	
Yes	159 (71.62)
No	63 (28.38)
**Metabolic Indices at Preconception**	
Homeostasis model assessment-insulin resistance (HOMA-IR) index, mean ± SD	1.27 ± 1.08
Triglycerides/high density lipoprotein cholesterol ratio	0.59 ± 0.41
Fatty liver index, mean ± SD	5.61 ± 10.38
Metabolic syndrome, *n* (%)	
Yes	7 (3.15)
No	215 (96.85)
**Prediabetes Status at Preconception**	
Impaired fasting glucose (IFG), *n* (%)	0 (0.00)
Impaired glucose tolerance (IGT), *n* (%)	11 (5.00)
Type 2 diabetes (T2D), *n* (%)	0 (0.00)
Normal glucose metabolism, *n* (%)	209 (95.00)
**OGTT at 24^+1^–28^+6^ Weeks’ Gestation**	
Glucose measures (mmol/L), mean ± SD	
Fasting glucose	4.28 ± 0.35
1-hour glucose	7.99 ± 1.52
2-hour glucose	6.68 ± 1.27
GDM, *n* (%)	
IADPSG/WHO 2013 criteria	29 (13.06)
**Adverse Birth Outcomes**	
Preterm birth, *n* (%)	
Yes	10 (4.50)
No	212 (95.50)
Low birthweight at term, *n* (%)	
Yes	7 (3.24)
No	209 (96.76)
Large for gestational age infant, *n* (%)	
Yes	34 (15.74)
No	182 (84.26)

**Table 2 ijerph-19-06792-t002:** Construction of preconception predictive risk model. The preconception predictive risk model for GDM was sequentially constructed using top predictors with SHAP value magnitudes greater than zero in the AutoML feature selection model. The optimal machine learning pipeline for each model and area under the receiver operating characteristic curve (AUC) performance metric are reported. The proposed AutoML model was also robust when replacing HbA_1c_ with fasting glucose (AUC: 0.87), replacing mean arterial blood pressure with systolic blood pressure (AUC: 0.91) and replacing fasting insulin with HOMA-IR index (AUC: 0.91) (Appendix A). HbA_1c_ had the greatest impact on model performance changes (ΔAUC = −0.06), followed by mean arterial blood pressure (ΔAUC = −0.02) and fasting insulin (ΔAUC = −0.02). Given these observations, maternal insulin resistance around conception can be postulated as an important determinant in the pathophysiology of metabolic diseases and fetal programming.

Features	Optimal Machine Learning Pipeline	AUC
1: HbA_1c_	Gradient boosting classifier.	0.81
2: HbA_1c_ + fatty liver index	Stacked ensemble model with logistic regression classifier, multinomial naïve Bayes classifier and multi-layer perceptron classifier.	0.78
3: HbA_1c_ + fatty liver index + mean arterial blood pressure	Stacked ensemble model with k-nearest neighbors classifier and decision tree classifier.	0.82
4: HbA_1c_ + fatty liver index + mean arterial blood pressure + fasting insulin	Stacked ensemble model with k-nearest neighbors classifier and decision tree classifier.	0.88
**5: HbA_1c_ + fatty liver index + mean arterial blood pressure + fasting insulin + TG/HDL ratio**	Extra trees classifier.	**0.93**
6: HbA_1c_ + fatty liver index + mean arterial blood pressure + fasting insulin + TG/HDL ratio + height	Stacked ensemble model with logistic regression classifier (stochastic gradient descent training) and k-nearest neighbors classifier.	0.89
7: HbA_1c_ + fatty liver index + mean arterial blood pressure + fasting insulin + TG/HDL ratio + height + age	Multi-layer perceptron classifier.	0.88
8: HbA_1c_ + fatty liver index + mean arterial blood pressure + fasting insulin + TG/HDL ratio + height + age + mid-upper arm circumference	Stacked ensemble model with Bernoulli naïve Bayes classifier, gaussian naïve Bayes classifier, multinomial naïve Bayes classifier and linear support vector machine classifier.	0.93
9: HbA_1c_ + fatty liver index + mean arterial blood pressure + fasting insulin + TG/HDL ratio + height + age + mid-upper arm circumference + BMI	Stacked ensemble model with extra trees classifier, Bernoulli naïve Bayes classifier and gaussian naïve Bayes classifier.	0.85
10: HbA_1c_ + fatty liver index + mean arterial blood pressure + fasting insulin + TG/HDL ratio + height + age + mid-upper arm circumference + BMI + parity	Stacked ensemble model with k-nearest neighbors classifier and multi-layer perceptron classifier.	0.85
11: HbA_1c_ + fatty liver index + mean arterial blood pressure + fasting insulin + TG/HDL ratio + height + age + mid-upper arm circumference + BMI + parity + alcohol consumption	Stacked ensemble model with gradient boosting classifier, multi-layer perceptron classifier and linear support vector machine classifier.	0.90
12: HbA_1c_ + fatty liver index + mean arterial blood pressure + fasting insulin + TG/HDL ratio + height + age + mid-upper arm circumference + BMI + parity + alcohol consumption + family history of diabetes mellitus	Stacked ensemble model with multinomial naïve Bayes classifier and multi-layer perceptron classifier.	0.87
13: HbA_1c_ + fatty liver index + mean arterial blood pressure + fasting insulin + TG/HDL ratio + height + age + mid-upper arm circumference + BMI + parity + alcohol consumption + family history of diabetes mellitus + Chinese ethnicity	Stacked ensemble model with multinomial naïve Bayes classifier and multi-layer perceptron classifier.	0.87
14: Mean arterial blood pressure + height + age + mid-upper arm circumference + BMI + parity + alcohol consumption + family history of diabetes mellitus + Chinese ethnicity	Stacked ensemble model with linear support vector machine classifier (stochastic gradient descent training), Bernoulli naïve Bayes classifier, multinomial naïve Bayes classifier, multi-layer perceptron classifier and linear support vector machine classifier.	0.81

**Table 3 ijerph-19-06792-t003:** Associations of top predictors and GDM outcome. Associations of top predictors identified from AutoML feature selection model and GDM outcome. Statistical tests were conducted two-sided with a significance level of 5%. All confidence intervals (CIs) are presented two-sided with a confidence level of 95%. The odds ratios (ORs) with 95% CI are presented. A resultant *p*-value of less than 0.05 is considered statistically significant.

Feature	GDM (*n* = 222)
	OR (95% CI) *p*-Value
HbA_1c_ (mmol/mol)	OR: 1.31 (1.12–1.53) *p*-value = 0.001 *
Fatty liver index	OR: 1.01 (0.98–1.05) *p*-value = 0.458
Mean arterial blood pressure (mm Hg)	OR: 0.99 (0.94–1.04) *p*-value = 0.584
Fasting insulin (mU/L)	OR: 1.05 (0.99–1.12) *p*-value = 0.119
Triglycerides/high density lipoprotein cholesterol ratio	OR: 1.45 (0.65–3.28) *p*-value = 0.365
Maternal height (cm)	OR: 0.96 (0.90–1.04) *p*-value = 0.311
Age (years)	OR: 0.97 (0.86–1.10) *p*-value = 0.673
Mid-upper arm circumference (cm)	OR: 1.05 (0.96–1.15) *p*-value = 0.290
BMI (kg/m^2^)	OR: 1.05 (0.97–1.13) *p*-value = 0.241
Parity	OR: 0.74 (0.32–1.71) *p*-value = 0.481
Self-reported alcohol consumption	OR: 2.06 (0.75–5.67) *p*-value = 0.161
Family history of diabetes mellitus	OR: 1.39 (0.61–3.18) *p*-value = 0.436
Chinese vs. Malay/Indian ethnicity	OR: 1.29 (0.47–3.60) *p*-value = 0.621
Feature	GDM (*n* = 222)
	OR (95% CI) *p*-value
HbA_1c_ (mmol/mol) ^	OR: 1.34 (1.13–1.60) *p*-value = 0.001 *
Feature	GDM (*n* = 211)
	OR (95% CI) *p*-value
HbA_1c_ (mmol/mol) ^#,^^	OR: 1.32 (1.10–1.59) *p*-value = 0.003 *

* Statistically significant feature. ^ Adjusted for maternal ethnicity, age, parity, family history of diabetes mellitus and pre-pregnancy BMI. ^#^ After excluding 11 women with prediabetes (impaired glucose tolerance) based on preconception OGTT.

**Table 4 ijerph-19-06792-t004:** Associations of top predictors and adverse birth outcomes (preterm birth, low birthweight at term and large for gestational age infant). Associations of top predictors identified from AutoML feature selection model and adverse birth outcomes (preterm birth, low birthweight at term and large for gestational age infant). Statistical tests were conducted two-sided with a significance level of 5%. All confidence intervals (CIs) are presented two-sided with a confidence level of 95%. The odds ratios (ORs) with 95% CI are presented. A resultant *p*-value of less than 0.05 is considered statistically significant.

Feature	Preterm Birth (*n* = 222)	Low Birthweight at Term (*n* = 216)	Large for Gestational Age Infant (*n* = 216)
	OR (95% CI) *p*-Value	OR (95% CI) *p*-Value	OR (95% CI) *p*-Value
HbA_1c_ (mmol/mol)	OR: 1.28 (1.01–1.62) *p*-value = 0.042 *	OR: 1.13 (0.86–1.49) *p*-value = 0.381	OR: 1.06 (0.92–1.21) *p*-value = 0.416
Fatty liver index	OR: 1.00 (0.94–1.06) *p*-value = 0.951	OR: 0.89 (0.68–1.16) *p*-value = 0.386	OR: 1.06 (1.03–1.10) *p*-value < 0.001 *
Mean arterial blood pressure (mm Hg)	OR: 1.02 (0.94–1.11) *p*-value = 0.688	OR: 0.96 (0.86–1.06) *p*-value = 0.403	OR: 1.03 (0.98–1.08) *p*-value = 0.253
Fasting insulin (mU/L)	OR: 1.04 (0.96–1.14) *p*-value = 0.317	OR: 1.05 (0.95–1.15) *p*-value = 0.359	OR: 1.08 (1.01–1.16) *p*-value = 0.019 *
Triglycerides/high density lipoprotein cholesterol ratio	OR: 0.79 (0.13–4.76) *p*-value = 0.797	OR: 1.42 (0.34–6.00) *p*-value = 0.630	OR: 2.85 (1.30–6.21) *p*-value = 0.009 *
Maternal height (cm)	OR: 0.95 (0.84–1.07) *p*-value = 0.363	OR: 0.91 (0.78–1.05) *p*-value = 0.192	OR: 0.99 (0.93–1.06) *p*-value = 0.794
Age (years)	OR: 1.05 (0.86–1.29) *p*-value = 0.629	OR: 0.97 (0.76–1.24) *p*-value = 0.799	OR: 1.07 (0.95–1.20) *p*-value = 0.267
Mid-upper arm circumference (cm)	OR: 0.97 (0.82–1.15) *p*-value = 0.737	OR: 0.90 (0.71–1.14) *p*-value = 0.362	OR: 1.22 (1.12–1.33) *p*-value < 0.001 *
BMI (kg/m^2^)	OR: 1.00 (0.88–1.16) *p*-value = 0.914	OR: 0.84 (0.64–1.10) *p*-value = 0.206	OR: 1.18 (1.09–1.27) *p*-value < 0.001 *
Parity	OR: 1.75 (0.49–6.25) *p*-value = 0.387	OR: 1.29 (0.28–5.90) *p*-value = 0.746	OR: 1.89 (0.90–3.95) *p*-value = 0.091
Self-reported alcohol consumption	OR: 0.38 (0.11–1.35) *p*-value = 0.134	OR: 2.47 (0.29–20.97) *p*-value = 0.407	OR: 0.44 (0.21–0.94) *p*-value = 0.033 *
Family history of diabetes mellitus	OR: 1.73 (0.47–6.35) *p*-value = 0.409	OR: 1.91 (0.41–8.78) *p*-value = 0.407	OR: 1.95 (0.92–4.17) *p*-value = 0.083
Chinese vs. Malay/Indian ethnicity	OR: 0.59 (0.15–2.39) *p*-value = 0.463	OR: 0.33 (0.07–1.51) *p*-value = 0.152	OR: 0.55 (0.24–1.26) *p*-value = 0.158
Feature	Preterm Birth (n = 185)		
	OR (95% CI) *p*-value		
HbA_1c_ (mmol/mol) ^	OR: 1.63 (1.12–2.38) *p*-value = 0.011 *		
Feature	Preterm Birth (*n* = 154)		
	OR (95% CI) *p*-value		
HbA_1c_ (mmol/mol) ^#,^^	OR: 1.75 (1.14–2.67) *p*-value = 0.010 *		
Feature	Large for Gestational Age Infant (*n* = 198)		
	OR (95% CI) *p*-value		
Fatty liver index ^	OR: 1.02 (0.96–1.08) *p*-value = 0.473		
Feature	Large for Gestational Age Infant (*n* = 198)		
	OR (95% CI) *p*-value		
Fasting insulin (mU/L) ^	OR: 1.01 (0.92–1.10) *p*-value = 0.825		
Feature	Large for Gestational Age Infant (*n* = 198)		
	OR (95% CI) *p*-value		
Triglycerides/high density lipoprotein cholesterol ratio ^	OR: 1.98 (0.76–5.10) *p*-value = 0.160		
Feature	Large for Gestational Age Infant (*n* = 198)		
	OR (95% CI) *p*-value		
Mid-upper arm circumference (cm) ^	OR: 1.21 (0.93–1.58) *p*-value = 0.162		
Feature	Large for Gestational Age Infant (*n* = 198)		
	OR (95% CI) *p*-value		
BMI (kg/m^2^) ^~^	OR: 1.20 (1.10–1.31) *p*-value < 0.001 *		
Feature	Large for Gestational Age Infant (n = 198)		
	OR (95% CI) *p*-value		
Self-reported alcohol consumption ^	OR: 0.47 (0.17–1.28) *p*-value = 0.138		

* Statistically significant feature. ^ Adjusted for maternal ethnicity, age, parity, family history of diabetes mellitus, pre-pregnancy BMI, GDM diagnosis, total gestational weight gain (derived by subtracting first antenatal visit weight from last antenatal visit weight) and child sex. ^~^ Adjusted for maternal ethnicity, age, parity, family history of diabetes mellitus, GDM diagnosis, total gestational weight gain (derived by subtracting first antenatal visit weight from last antenatal visit weight) and child sex. ^#^ After excluding 11 women with prediabetes (impaired glucose tolerance) based on preconception OGTT.

## Data Availability

The data supporting the findings of this research can be requested from the S-PRESTO executive committee upon reasonable request. The code generated to reproduce this research is available at GitHub page: https://github.com/mukkeshkumar/S-PRESTO_Gestational-Diabetes-Mellitus. The AutoML model has been deployed as a web application and can be accessed through the following URL: https://www.mornin-feng.com/all-projects-and-demos#gdm3.

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
