# Peer review of "Automated Machine Learning (AutoML)-Derived Preconception Predictive Risk Model to Guide Early Intervention for Gestational Diabetes Mellitus"

_ijerph, 2022, doi:10.3390/ijerph19116792_

Round 1

Reviewer 1 Report

IJERPH 1664926 Automated Machine Learning (AutoML) derived preconception predictive risk model to guide early intervention for gestational diabetes mellitus

This is an interesting manuscript identifying a relatively small number of factors that can predict the risk of GDM preconceptionally. The organisation of the results can be much improved and the implications should be discussed more broadly.

Major comments:

  1. Results, figure 2: was the increase in bodyweight at 3 months post-partum significantly different e.g. both statistically and clinically? The text in the figure is so small that I could not read it (when zooming, resolution was lost). The same question applies to the other components of figure 2.
  2. Similarly, are the results presented in figure 3, statistically and/or clinically significant.
  3. Figure 4 is not very meaningful and can be moved to the supplementary material with the change in the proportion or number of participants who smoked or drank presented in text format instead. Fig 5d is similarly not very informative.
  4. The results are presented for the full cohort, to what extent is there variability in the patterns between individual participants in the groups? Would it be more meaningful to present the data as a proportional change from the pre-conception value? This may get around the large variability in individual values for each component that is visible in figures 2, 3 and 5.
  5. Results, factors included in the model: given that BMI is calculated using height, is there collinearity for these two factors and if so, does that affect the modelling? Similarly, is there collinearity between Fatty Liver Index and the TG/HDL ratio?
  6. Table 2b: I am not sure to what extent this needs to be shown in this much detail, perhaps move to the supplementary material?
  7. Tables 3a-c could be presented in a single table instead as could tables 4a-c
  8. There is no definition for preterm birth and low birthweight: please define these. Is there any chance that some of the preterm birth for the people with high HbA1c have been delivered iatrogenically?
  9. The results show that the predictive power of the algorithm is excellent (both the one based on blood markers and the one based on non-invasive measures). To what extent do women in Asia attend pre-conception counselling and would the model also work in early pregnancy to identify women at high risk for developing GDM?
  10. Is there any data that interventions work before or during pregnancy that successfully lower HbA1c; TG/HDL ratio; mean arterial blood pressure, fasting glucose or are these factors mostly determined by genetics and therefore difficult to sway?

Minor comments:

  1. In table 1, there is a partial misalignment of the variables and the values, please fix.
  2. Figure 1 is mislabeled, this seems to be the CONSORT diagram.
  3. Figure 3-5 are not fully present in the file. Figure legends are missing for most figures.
  4. HbA1c is presented as % in table 1 but mmol/mol in tables 3-4. Could this be made consistent?

Reviewer 2 Report

This is a comprehensive study of predictors for gestational DM using advanced AUC machine learning methods. The presentation of the results is logically formulated and explained.  The limitations of the study are carefully documented.  One suggestion is that the interaction effect of multiple predictors could be incorporated into the assessment of the predictive power of the risk model.  For instance, body weight may exert some influence on the variation in odds ratio.  In other words, the authors could explore the possible synergistic effect of environmental and personal factors on gestational DM and its outcomes.

Reviewer 3 Report

The manuscript is devoted to an Automated Machine Learning model to predict GDM risk based on pre-conception parameters. The ability to predict GDM with a high accuracy already at pre-conception is the strength of the study. However, there are some issues pointed below.

  • Please provide the formula including coefficients for the best predictive model. Otherwise, the accuracy of the model can not be verified by external validation.
  • What is the difference between table 2a and table 2b? In both tables «The optimal machine learning pipeline for each model and area under the receiver operating characteristic curve (AUC) performance metric are reported».
  • The part concerning Visualization of Trends in Longitudinal Data seems redundant and can be removed. It confirms well known facts and is not necessary to fulfill the aim of the study.
  • Table 1 contains a systemic error with numbers shifted from their original place (starting from Prediabetes Status at Preconception)
  • Why did you choose to show only Preterm Birth and Low Birthweight at Term among Adverse Birth Outcomes? It would be logical to add at least LGA infants.
  • Please tone down the claims of a «revolutionary» model. It needs to be externally validated at least. The phrase in the introduction «We therefore evaluated a paradigm shift in GDM management strategy by developing a preconception-based predictive care model …» sounds a bit misleading. The study did not evaluate the results of the implementation of the developed model. A word like «suggest» would be more appropriate instead of «evaluated».
  • I would suggest improving the introduction by discussing at least briefly different options to prevent GDM.
  • The authors refer only to NICE guidelines concerning GDM screening. However, worldwide there are different ways to screen and diagnose GDM.
  • The part of the discussion concerning Comparison with Prior Work is not sufficient. There are numerous publications on different models for GDM prediction in early pregnancy (eg, PMID: 35073990), including a few models with similar accuracy (an AUC of >0,9) (eg., PMID: 32762275).
